# A Nested Chinese Restaurant Topic Model for Short Texts with Document Embeddings

**Yue Niu** **, Hongjie Zhang** and **Jing Li** *

Department of Computer Science and Technology, University of Science and Technology of China, Hefei 230052, China; nyustc36@mail.ustc.edu.cn (Y.N.); zhanghongjie@mail.ustc.edu.cn (H.Z.)
* Correspondence: lj@ustc.edu.cn

**Abstract:** In recent years, short texts have become a kind of prevalent text on the internet. Due to the short length of each text, conventional topic models for short texts suffer from the sparsity of word co-occurrence information. Researchers have proposed different kinds of customized topic models for short texts by providing additional word co-occurrence information. However, these models cannot incorporate sufficient semantic word co-occurrence information and may bring additional noisy information. To address these issues, we propose a self-aggregated topic model incorporating document embeddings. Aggregating short texts into long documents according to document embeddings can provide sufficient word co-occurrence information and avoid incorporating non-semantic word co-occurrence information. However, document embeddings of short texts contain a lot of noisy information resulting from the sparsity of word co-occurrence information. So we discard noisy information by changing the document embeddings into global and local semantic information. The global semantic information is the similarity probability distribution on the entire dataset and the local semantic information is the distances of similar short texts. Then we adopt a nested Chinese restaurant process to incorporate these two kinds of information. Finally, we compare our model to several state-of-the-art models on four real-world short texts corpus. The experiment results show that our model achieves better performances in terms of topic coherence and classification accuracy.

**Keywords:** topic model; text mining; document embeddings; short text

## 1. Introduction

With the growth of social media and applications of mobile phones, short texts have been a kind of prevalent and important information on the internet. There is abundant semantic information that can be found in short texts. However, unlike documents with regular size, the average length of each short text is very short. This characteristic makes the knowledge discovery on short texts being a challenging research problem.

To discover knowledge from texts, the topic modeling method is widely used [1]. For documents with regular size, conventional topic models like LDA [2] and HDP [3] perform well. These methods can automatically generate topics according to word co-occurrence information. However, for short texts, word co-occurrence information is very sparse because of the short length of each text [4]. So the performances of these conventional methods are very poor on short texts [5]. To overcome this problem, many researchers propose different kinds of customized topic models dealing with the sparsity of word co-occurrence information [6].

As word co-occurrence information in short texts is sparse, researchers propose two kinds of strategies to overcome this problem. The first strategy is to incorporate auxiliary information. Metadata like hashtags or authors is available in short texts. Some methods aggregate short texts according to metadata [7–9]. However, these methods are effective only for a few short texts corpus because metadata is not always available. So other methods incorporate word embeddings generated from an auxiliary corpus with documents

of regular length [10–14]. However, these methods need to construct a semantically corresponding auxiliary corpus of short texts. If the auxiliary corpus is not semantically related to target short texts, the performance will be very poor [15].

The second strategy is to design specific models without auxiliary information. Several models restrict the number of topics each short text corresponds to [16–18]. The local word co-occurrence information of each topic is very sparse. So when a short text corresponds to one or a few topics, the local word co-occurrence information will be less sparse. However, this strategy cannot add sufficient word co-occurrence information. Several models use the global word co-occurrence information [19,20]. These methods rebuild short texts corpus and construct a global word pairs set. The word pairs set is the summary of local word co-occurrence information. So the global word co-occurrence information will be less sparse. However, in these models, variables of documents are missing and the probability distribution between short texts and topics is unable to be obtained directly. Self-aggregated topic models [21,22] automatically aggregate short texts into latent long documents. Then topics are generated from these long documents. Long documents incorporate sufficient word co-occurrence information and make local word co-occurrence information no longer sparse. These models seem more reasonable than other strategies and auxiliary information is not needed. However, the aggregation process of short texts may incorporate non-semantic information. Short texts that are not semantically relevant may be aggregated into one long document. Then this long document will bring a lot of non-semantic word co-occurrence information into topics and makes topics incoherent.

As there exist a lot of deficiencies of the state-of-the-art topic models, we propose a self-aggregated topic model by incorporating document embeddings [23] (DESTM) to deal with the sparsity of word co-occurrence information. Document embedding information of short texts is the mapping of short texts' content in vector space. This information contains two kinds of information: word co-occurrence information and context semantic information. Then we can get the similarities between every two short texts according to document embeddings. So, if short texts are aggregated according to the similarities calculated from document embeddings, semantically related short texts are easier to be aggregated together. Long documents generated by our model can effectively avoid incorporating non-semantic word co-occurrence information.

However, document embeddings of short texts also contain a lot of noisy information resulting from the sparsity of word co-occurrence information. So the challenge is that we need to exclude noisy information in document embeddings. Noisy information affects the value of distances between short texts and makes short texts not similar. So we changed document embeddings into global and local semantic information to discard noisy information and retain as much reliable semantic information as possible.

The global semantic information is the probability distribution of the content similarities on the entire dataset. Although the distances of similarities are affected by the noisy information, the distribution of the entire dataset will not change. For example, if the distribution of similarities is a normal distribution, noisy information cannot change this distribution to other kinds of distributions. However, the noisy information may have little impact on the parameter of the similarity distribution. So we only incorporate this distribution as the prior in order to minimize the impact of its parameters.

The local semantic information is the distances between similar short texts. The sparsity of word co-occurrence information will reduce the similarities between short texts. So we use a threshold to retain distances between short texts that are relatively similar. Distances between short texts that are not so similar are discarded.

Then we adopt the nested Chinese restaurant process to incorporate global and local semantic information in two steps. In the first step, we generate the distribution of similarities of short texts by a Chinese restaurant process. In the second step, we incorporate this distribution as the prior distribution and generate long documents according to the distances of similar short texts. Contributions of our model are as follows:

- Firstly, our model aggregates short texts into latent long documents according to document embeddings. Document embedding information provides similarities of short texts and can avoid incorporating non-semantic word co-occurrence information.
- Secondly, we change document embeddings into global and local semantic information to discard noisy information. Global information is the similarity probability distribution across all short texts and local information is the distances of similar short texts. Our model adopts the nested Chinese restaurant process to incorporate these two kinds of information in two steps.
- Thirdly, we compare our model to several state-of-the-art models. By generating topics on four real-world short texts corpus, the experimental results demonstrate that our model outperforms other methods in terms of topic coherence and classification accuracy.

The remainder of this paper is organized as follows. In Section 2, we analyze the state-of-the-art methods of topic models for short texts. In Section 3, we introduce the details of our model. In Section 4, we show experiments and analyze results. In Section 5, we conclude our work and discuss future works.

## 2. Related Works

There are mainly two kinds of strategies. One is incorporating auxiliary information to provide additional word co-occurrence information. Another is designing specific models without auxiliary information.

### 2.1. Models with Auxiliary Information

For the first strategy, one kind of auxiliary information is the metadata of short texts. The AT model [7] aggregates short texts according to the author of texts. ET-LDA [24] generates topics from events and each event is constructed by associated tweets. The pooling model [8] aggregates tweets according to hashtags. Other models combine different kinds of metadata. mLDA [9] aggregates short texts according to authors and labels. rrPLSA [25] aggregates short texts according to authors and social roles. The AOTM model [26] incorporates authors and authors' regular sized documents. However, for different kinds of short texts, metadata may not always be available. Another kind of auxiliary information is the corpus of documents with regular size. Word embeddings [27,28] are always used to incorporate semantic information from the long document set into topic models. The models tf-lda and tf-dmm [10] generate words from topics by a multi-nominal distribution or by feature vectors from word embeddings. The Gaussian LDA model [11] proposes a multivariate Gaussian distribution between topics and word vectors instead of the multinominal distribution. The GPU-DMM model [12] incorporates word embedding through the Pólya Urn strategy. The model GLTM [13] combines both local word embeddings from short texts and global word embeddings from auxiliary long documents. The TRNMF [29] model incorporates word embeddings based on regularized non-negative matrix factorization. The TSSE-DMM [30] model divides topics into four aspects and generates topics along with word embeddings. The WEPTM model [14] generates priors of the distribution between topics and words according to word embeddings. However, if the auxiliary document set is not carefully designed and becomes semantically irrelevant with target short texts, word embeddings will be noisy information and lead to poor performance [15].

### 2.2. Models without Auxiliary Information

The second strategy is to design specific models without auxiliary information. The GSDMM model [16] suggests that each short text only corresponds to one topic. The PDMM model [17] uses a Poisson distribution to limit the number of topics for each short text. The DSPTM model [18] restricts the number of topics for each short text by the "Spike and Slab" prior. All these models can improve the problem of the sparsity of local word co-occurrence information, but can not provide sufficient additional word co-occurrence information. The BTM model [19] rebuilds short texts into word pairs according to global

word co-occurrence information. The WNTM model [20] transforms short texts into a network of words according to global word co-occurrence information. These two methods use global word co-occurrence information but cannot provide distributions between topic and short texts directly. The SATM model [21] proposes a self-aggregated topic model. This model uses two phases sampling long documents and sampling topics. However, it is time-consuming and may become overfitting. The PTM model [22] proposes a model heuristically generating latent long documents and generating topics from these long documents. This model is more simple and effective compared to SATM model. The SPTM model [22] avoids long documents being ambiguous when the number of long documents is defined as too small. Although all the self-aggregated models provide sufficient additional word co-occurrence information, the aggregation process will incorporate non-semantic word co-occurrence information inevitably.

The classification of the state-of-the-art topic models for short texts are shown in Table 1 according to their strategies.

**Table 1.** Classification of topic models for short texts.

| Strategy | Models |
|---|---|
| Models with auxiliary data | Incorporating metadata [7–9,24–26] Incorporating a long document set [10–14,29,30] |
| Models without auxiliary data | Limiting topics correspond to each text [16–18] Using global word co-occurrence information [19,20] Aggregating short texts automatically [21,22] |

## 3. Model and Inference

In this section, we introduce the details of our model, DESTM. Firstly, we introduce the generation process and the probabilistic graphical model. Then we show how we incorporate document embeddings through the nested Chinese restaurant process. Then we show the inference equations and the method for sampling latent topics and latent long documents according to Gibbs sampling.

### 3.1. Overview

Topic modeling methods sample topics through a joint probability distribution constructed by a generation process. We chose the conventional method LDA as an example. The LDA method designs three variables: text $d$, word $w$, and topic $z$. Then model constructs a generation process among these variables. It suggests that for each word $w$ in each text $d$, a topic $z$ is sampled from text $d$ according to a multi-nominal distribution, and word $w$ is sampled from the topic $z$ according to another multi-nominal distribution. Then the joint probability distribution $P(d,z,w)$ is define by two multi-nominal distributions $P(z|d)$ and $P(w|z)$, where $P(d,z,w) = P(w|z)P(z|d)$. The two variables $w$ and $d$ are already known and variable $z$ is the latent variable. So we can sample $z$ according to the Gibbs sampling method [31].

To deal with short texts, self-aggregated topic models design a new latent variable $l$ as the long document. The state-of-the-art model PTM proposes a generation process among four variables: short text $d$, word $w$, topic $z$, and long document $l$. Firstly, for each short text $d$, sample a long document $l$ according to a multi-nominal distribution. Then for each word $w$ of short text $d$, sample a topic $z$ from a long document $l$ according to a multi-nominal distribution. Finally, sample the word $w$ from topic $z$ according to another multi-nominal distribution. So, this generation process constructs a joint probability distribution $P(d,w,z,l)$ according to three multi-nominal distributions $P(w|z)$, $P(z|l)$ and $P(l|d)$, where $P(d,w,z,l) = P(w|z)P(z|l)P(l|d)$. Every short text shares a same multi-nominal distribution $P(l|d)$. So if the number of long documents is less than the number of short texts, short texts will be aggregated into long documents.

However, the aggregation according to a multi-nominal distribution cannot avoid aggregating semantically irrelevant short texts into one long document. So we propose a nested Chinese restaurant process to incorporate document embeddings instead of using a multi-nominal distribution. As the Chinese restaurant process is a specific sampling method of the Dirichlet process. So firstly, we use the Dirichlet process (DP) to describe the generalized mathematical representation of the generation process of our model. Then we will introduce details of the nested Chinese restaurant process in our model.

The generation process is as follows:

1. Sample $G_0 \sim DP(\alpha, H)$
2. Sample $G \sim DP(\delta, G_0)$
3. For each topic $z$
   Sample $\eta_z \sim Dir(\gamma)$
4. For each latent long document $l$
   Sample $\theta_l \sim Dir(\beta)$
5. For each short text $i$

   (a) Sample a long document $l \sim G$
   (b) For each word $w_{i,l}$ in short text $i_l$

      i. Sample a topic $z \sim Multi(\theta_l)$
      ii. Sample the word $w_{i,l} \sim Multi(\eta_z)$

The Dirichlet process is a stochastic process that samples distributions. In our generation process, we first sample a distribution $G_0$ from a Dirichlet process $DP(\alpha, H)$, where $\alpha$ is the prior parameter and $H$ is the prior distribution. Then using $G_0$ as another prior distribution, we sample a distribution $G$ from a Dirichlet process $DP(\delta, G_0)$, where $\delta$ is the prior parameter. Therefore, the expectation of $G$ is $G_0$. Then, for each short text $i$, we sample a long document $l$ according to distribution $G$. Here, $G$ can be regarded as a probability measure that divides the space of short texts. So, short texts are aggregated into long documents according to $G$. Then, for each word $w_{i,l}$ in short text $i_l$ aggregated to long document $l$, we firstly sample a topic $z$ from a multi-nominal distribution $Multi(\theta_l)$. The parameter $\theta_l$ belongs to the long document $l$. Secondly, we sample the word $w_{i,l}$ from another multi-nominal distribution $Multi(\eta_z)$. The parameter $\eta_z$ belongs to the topic $z$.

Through this generation process, we build the joint probabilistic distribution $P(w, d, z, l) = P(w|z)P(z|l)GG_0$. The relations between these variables are shown in the graphical model in Figure 1.

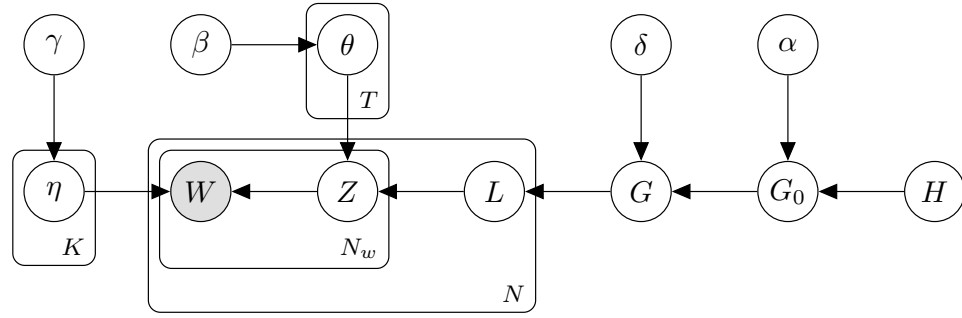

**Figure 1.** DESTM graphical model.

As shown in the graphical model. $N$ means the number of short texts. $N_w$ means the number of words in a short text. $T$ means the number of topics and $K$ means the number of vocabulary. Solid node $W$ means words that are known variables. Nodes $Z$ and $L$ are the latent variables that we need to sample. Nodes $\gamma$, $\beta$, $\delta$, $\alpha$, and $H$ are prior parameters that we need to define. Nodes $\theta$ and $\eta$ are parameters of multi-nominal distributions. $G_0$ is a probability measure sampled from the Dirichlet process and a probability measure $G$ is sampled according to prior $G_0$.

In our model, we use the nested Chinese restaurant process to sample $G_0$ and $G$. Document embeddings are separated into two kinds of semantic information. One is the distribution of similarities of short texts and another is the distances of similar short texts. Then we incorporate the distribution into $G_0$ and the distances into $G$. As the expectation of $G$ is $G_0$, $G$ will contain both two kinds of semantic information. By aggregating short texts according to the distribution $G$, short texts that are semantically related are more likely to be aggregated into one long document and can significantly avoid incorporating non-semantic word co-occurrence information.

### 3.2. Nested Chinese Restaurant Process

In our model, we use the nested Chinese restaurant process to sample $G_0$ and $G$ from two Dirichlet processes. The nested Chinese restaurant process can be described as follows:

1. Produce a Chinese restaurant $r$ with unlimited number of tables
2. The first customer sits at the first table
3. The customer $c$ sits at
   - (a) The table $k$ with probability $\frac{n_k}{\alpha_0 + n - 1}$
   - (b) A new table with probability $\frac{\alpha_0}{\alpha_0 + n - 1}$
4. For each table $k$ with customers $\mathbf{c}$ in restaurant $r$
   - (a) Produce a Chinese restaurant $r_k$ with unlimited number of tables
   - (b) The first customer in $\mathbf{c}$ sits at the first table
   - (c) The customer $c_k$ in $\mathbf{c}$ sits at
     - i. The table $k^*$ with probability $\frac{n_{k^*}}{\alpha_0^* + n_k - 1}$
     - ii. A new table with probability $\frac{\alpha_0^*}{\alpha_0^* + n_k - 1}$

Following this procedure, tables are sampled by customers. $r$ means the Chinese restaurant for all customers. $n_k$ means the number of customers of table $k$. $n$ means the number of total customers in the restaurant. $\alpha_0$ is the discrete parameter that determines the dispersion of customers. $\mathbf{c}$ means a subset of customers. $r_k$ means the Chinese restaurant for customers in table $k$. $n_{k^*}$ means the number of customers sit at table $k^*$ in restaurant $r_k$. $\alpha_0^*$ is the discrete parameter that determines the dispersion of customers $\mathbf{c}$.

A nested Chinese restaurant process is a two-step Chinese restaurant process. In the first step, a Chinese restaurant process assigns customers to $K$ tables. Then, for each table $k$ in this table set, we can get a subset of customers $\mathbf{c}$ that sits at the table $k$. So in the second step, for each customer $c_K \in \mathbf{c}$, a new Chinese restaurant process assigns customers to $K^*$ tables. Finally, for all tables generated from the first step, we generate $K$th new Chinese restaurant processes in the second step. In these two steps, the first step samples a distribution $G_0$. The second step incorporates $G_0$ as the prior and generates a new distribution $G$. As the expectation of $G$ is $G_0$, customers will generate tables in the second step following the distribution $G_0$. In our model, short texts are customers and long documents are tables. In the next section, we will introduce how document embeddings are incorporated into our model.

### 3.3. Incorporating Document Embeddings

Document embedding information is calculated from short texts accord to the PV-DM method [23]. Each short text is mapped to a vector. Then we calculate the cosine distance of every two short texts. Suppose the embeddings vector of the short text $a$ is $\mathbf{A}$ and the short text $b$ is $\mathbf{B}$. The cosine distance $D(a, b)$ between these two short texts is calculated as follows:

$$D(a, b) = \frac{\mathbf{A} \cdot \mathbf{B}}{||\mathbf{A}|| \, ||\mathbf{B}||} \tag{1}$$

The cosine distance $D(a, b)$ is also the similarity distance of short texts $a$ and $b$. Then we incorporate the cosine distance of every two short texts into the nested Chinese restaurant process in our model.

In the first step of the nested Chinese restaurant process, we incorporate all cosine distances. Then in the second step of the nested Chinese restaurant process, we incorporate distances between similar short texts. By designing a threshold, distances of short texts that are not similar are discarded. The filtering process is as follows:

1.　If $D(a, b) \geq \eta$, then $D_s(a, b) = D(a, b)$
2.　Else $D(a, b) \leq \eta$, then $D_s(a, b) = 0$

Here, $\eta$ is the threshold. $D(a, b)$ is the cosine distance. Because the more similar short text $a$ and $b$ are, the greater the value of $D(a, b)$ will be. So threshold $\eta$ can filter the cosine distances and retains distances of similar short texts that are greater than the value of $\eta$. Finally, we generate the filtered distance $D_s(a, b)$

Then, we generate the nested Chinese restaurant process according to the cosine distances. The generation process is as follows:

1.　The first short texts samples the first long document
2.　For each short text $i$

　(a)　For each long document $l^*$
　　　$D(i, l^*) = \sum D(i, i_{l*})$

　(a)　Sampling long document $l^*$ with probability $\frac{n_{l*} + D(i, l^*)}{\alpha^* + n - 1}$
　(b)　Sampling a new document with probability $\frac{\alpha^*}{\alpha^* + n - 1}$

3.　For each long document $l^*$ with short texts set $I_{l*}$

　(a)　The first short texts in $I_{l*}$ samples the first long document
　(b)　For each short text $i \in I_{l*}$

　　i.　For each long document $l$
　　　$D_s(i, l) = \sum D_s(i, i_l)$

　　i.　Sampling long document $l$ with probability $\frac{n_l + D_s(i, l)}{\alpha + n_{l*} - 1}$
　　ii.　Sampling a new document with probability $\frac{\alpha}{\alpha + n_{l*} - 1}$

In the first step of the nested Chinese restaurant process, $D(i, i_{l*})$ means the cosine distance between short text $i$ and short text $i_{l*}$ in long document $l^*$. Then, $D(i, l^*)$ is the summary of cosine distances between short text $i$ and every short text in long document $l^*$. $n_{l*}$ is the number of short texts in long document $l^*$. $\alpha^*$ is the discrete parameter that determines the dispersion of short texts. In the second step of the nested Chinese restaurant process, $D_s(i, i_l)$ is the cosine distance between similar short texts. $D_s(i, l)$ is the summary of distances between short text $i$ and all similar short texts in document $l$. $n_l$ is the number of short texts in long document $l$ and $\alpha$ is the discrete parameter that determines the dispersion of short texts in $l^*$.

After this nested Chinese restaurant process, we finally sample a long document $l$ for each short text. Each long document is generated according to the global and local semantic information. The global semantic information is the distribution of the similarities of short texts generated through the first step of the nested Chinese restaurant process. The local semantic information is the cosine distances between similar short texts. Then the global semantic information becomes the prior of the second step of the nested Chinese restaurant process. Every long document $l$ is sampled according to these two kinds of semantic information.

## 4. Inference

We train our model according to the collapsed Gibbs sampling method. The joint probability distribution of our model is $p(l^*, l, z, w, \alpha, \beta, \gamma, \delta)$, where $l^*$ is the long document variable generated from the first step of the nested Chinese restaurant process, $l$ is the long document variable generated from the second step of the nested Chinese restaurant

process, $z$ is the variable of the topic, $w$ is the variable of the word, $\alpha$ is the dispersion prior of short texts that sample $l^*$, $\beta$ is the dispersion prior of short texts that sample $l$, $\gamma$ is the prior of multi-nominal distribution between $z$ and $l$, and $\delta$ is the prior of multi-nominal distribution between $z$ and $w$. By integrating out parameters in this distribution, we sample three latent variables: the long document $l^*$ generated in the first step of the nested Chinese restaurant process, the long document $l$ in the second step, and latent topics.

*4.1. Sampling Long Documents Assignments $l^*$*

Samples of $l^*$ are generated from a Chinese restaurant process. For the $i$th short text in short texts corpus, the probability $\phi_i$ of sampling long document $l^*$ is as follows:

$$\phi_i | \phi_1, ..., \phi_{i-1}, \alpha, H \sim \sum_{l^*=1}^{L^*} \frac{n_{l^*}}{\alpha + n - 1} \delta_{\theta_l^*} + \frac{\alpha}{\alpha + n - 1} H \tag{2}$$

Here $H$ is the prior distribution. $L^*$ is the number of long document samples of $l^*$. $n_{l^*}$ is the number of short texts in long document $l^*$. $n$ is the total number of short texts in the dataset. $\delta_{\theta_{l^*}}$ means the atom of long document $l^*$. The sequence of short text is exchangeable in the Chinese restaurant process. So when sampling long document $l^*$ from any short text $i$, we suggest that $i$ is the last short text in corpus. For each short text $i$, the probability function of sampling long document $l^*$ is designed as follows:

$$p(l_i^* = l^* | l_{-i}^*, l, w, z, \alpha, \beta, \gamma, \delta) \propto \frac{p(l_i^* | \alpha)}{p(l_{-i}^* | \alpha)} \tag{3}$$

$$\propto \begin{cases} n_{l^*} + d(i, l^*), & l^* <= L^* \\ \alpha, & l^* = L^* + 1 \end{cases} \tag{4}$$

Here, $l_i^*$ means the long document that short text $i$ sampled. $l_{-i}^*$ means the long document excluding short text $i$. $n_k^*$ means the number of short texts in long document $k$. $d(i, k^*)$ is the summary of cosine distances. We calculate $d(i, l^*)$ as $d(i, l^*) = \sum d(i, j)$. $j$ is the short text in long document $k$ and $d(i, j)$ is the cosine distance between short text $i$ and $j$.

Finally, we generate the probability of sampling $l^*$ as:

$$p(l_i^* = l^* | l_{-i}^*, l, w, z, \alpha, \beta, \gamma, \delta) \propto \begin{cases} n_{l^*} + d(i, l^*), & l^* <= L^* \\ \alpha, & l^* = L^* + 1 \end{cases} \tag{5}$$

*4.2. Sampling Long Documents Assignments $l$*

After sampling long document $l^*$, we will sample long documents $l$. Samples of $l$ are also generated from a Chinese restaurant process. Each long document $l^*$ generates a short texts set $I_{l^*}$. For each short text $i \in I_{l^*}$, we aggregate short texts according to another Chinese restaurant process. The probability $\phi_i$ of sampling long document $l$ is as follows:

$$\phi_i | \phi_1, ..., \phi_{i-1}, \beta, H_{l^*} \sim \sum_{l=1}^{L} \frac{n_l}{\beta + n_{l^*} - 1} \delta_{\theta_l} + \frac{\beta}{\beta + n_{l^*} - 1} H_{l^*} \tag{6}$$

Here $H_{l^*}$ is the prior distribution generated by $l^*$. $L$ is the number of long document samples of $l$. $n_l$ is the number of short texts assigned to $l$. $\delta_{\theta_l}$ means the atom of long document $l$. For each short text $i \in I_{l^*}$, we also suggest that $i$ is the last short text in $I_{l^*}$. The probability function of sampling long document $l$ is designed as follows:

$$p(l_i = l | l_{-i}, l^*, w, z, \alpha, \beta, \gamma, \delta) \propto \frac{p(l_i | \beta)}{p(l_{-i} | \beta)} \frac{p(w, z | l_i, \beta, \gamma, \delta)}{p(w, z | l_{-i}, \gamma, \delta)} \tag{7}$$

Here $l_i$ means the long document sample of short text $i$. $l_{-i}$ means the long document excluding short text $i$. The equation contains two probabilities, we will generate them separately.

Firstly, we generate the probability $\frac{p(l_i|\beta)}{p(l_{-i}|\beta)}$ as follows:

$$\frac{p(l_i|\beta)}{p(l_{-i}|\beta)} \propto \begin{cases} n_l, & l <= L \\ \beta, & l = L+1 \end{cases} \tag{8}$$

Secondly, we generate the probability $\frac{p(w,z|l_i,\beta,\gamma,\delta)}{p(w,z|l_{-i},\gamma,\delta)}$. This probability can be simplified as follows:

$$\frac{p(w,z|l_i,\gamma,\delta)}{p(w,z|l_{-i},\gamma,\delta)} \propto \frac{p(z|l_i,\gamma)}{p(z_{-i}|l_{-i},\gamma)} \tag{9}$$

Here $z_{-i}$ means the topics that excluding topics in short text $i$.

If long document $l$ is sampled from an existing long document, we generate the probability $\frac{p(z|l_i,\gamma)}{p(z_{-i}|l_{-i},\gamma)}$ as follows:

$$\frac{p(z|l_i,\gamma)}{p(z_{-i}|l_{-i},\gamma)} \propto \frac{\prod^{t \in i} \prod_{j=1}^{n_{i,t}}(n_{l,t}^{-i} + \gamma + j - 1)}{\prod_{g=1}^{n_i}(n_l^{-i} + T\gamma + g - 1)} \tag{10}$$

Here $t \in i$ means topics sampled from short text $i$. $n_{i,t}$ means the number of topic $t$ in short text $i$. $n_{l,t}^{-i}$ means the number of topic $t$ in document $l$ excluding counts of topic $t$ in short text $i$. $n_i$ means the length of short text $i$. $n_l^{-i}$ means the length of long document $l$ excluding short text $i$. $T$ means the total number of short texts.

If long document $l$ is sampled from a new document, we generate the probability $\frac{p(z|l_i,\gamma)}{p(z_{-i}|l_{-i},\gamma)}$ as follows:

$$\frac{p(z|l_i,\gamma)}{p(z_{-i}|l_{-i},\gamma)} \propto \frac{\prod^{t \in i} \prod_{j=1}^{n_{i,t}}(\gamma + j - 1)}{\prod_{g=1}^{n_i}(T\gamma + g - 1)} \tag{11}$$

Finally, we generate the sampling probability of long document $l$ as follows:

$$p(l_i = l|l_{-i}, l^*, w, z, \alpha, \beta, \gamma, \delta) \propto \begin{cases} n_l \frac{\prod^{t \in i} \prod_{j=1}^{n_{i,t}}(n_{l,t}^{-i} + \gamma - j)}{\prod_{g=1}^{n_i}(n_l^{-i} + T\gamma - g)}, & l <= L \\ \beta \frac{\prod^{t \in i} \prod_{j=1}^{n_{i,t}}(\gamma + j - 1)}{\prod_{g=1}^{n_i}(T\gamma + g - 1)}, & l = L+1 \end{cases} \tag{12}$$

*4.3. Sampling Topics Assignments z*

After sampling long documents $l$, we generate samples of topic $z$. For each word $w$ in each short text $i$, we generate the probability of sampling topic $z$ as follows:

$$p(z_l = z|z_{-w}, w, l, l^*, \alpha, \gamma, \delta) \propto \frac{p(z_l|l,\beta)}{p(z_{-w}|l,\beta)} \frac{p(w|z,\gamma)}{p(w|z_{-w},\gamma)} \tag{13}$$

Here $z_l$ means topic $z$ belongs to long document $l$. $z_{-w}$ means topics in corpus excluding the topic of word $w$. For this equation, we first generate the probability of $\frac{p(z_l|l,\beta)}{p(z_{-w}|l,\beta)}$ as follows:

$$\frac{p(z_l|l,\gamma)}{p(z_{-l}|l,\gamma)} \propto n_{l,z} + \gamma - 1 \tag{14}$$

$n_{l,z}$ means the number of topics in long document $l$.

Then we generate the probability of $\frac{p(w|z,\gamma)}{p(w|z_{-w},\gamma)}$ as follows:

$$\frac{p(w|z,\delta)}{p(w|z_{-l},\delta)} \propto \frac{n_{w,z}^{-w} + \delta - 1}{n_z^{-w} + V\delta - 1} \tag{15}$$

Here $n_{w,z}^{-w}$ means the number of the same word $w$ of topic $z$ excluding the topic of word $w$. $n_z^{-w}$ means the number of total words of topic $z$ excluding word $w$. $V$ means the length of vocabulary. Finally, we obtain the probability of sampling topic $z$ as follows:

$$p(z_l = z | z_{-l}, w, l, l^*, \beta, \gamma, \delta) \propto (n_{l,z} + \gamma - 1) \frac{n_{w,z}^{-w} + \delta - 1}{n_z^{-w} + V\delta - 1} \tag{16}$$

*4.4. DESTM Gibbs Sampling Process*

After generating probabilities of sampling, we show the detailed Gibbs sampling process in Algorithm 1. The conventional iteration of the Gibbs sampling method is more than one thousand. In one iteration, for each short text, we first sample long document $l^*$ as the prior and long document $l$ as the aggregation of short texts. Then for each short text with a sample of $l$, we sample the topic $z$ of each word in the topic.

## 5. Experimental Results

In this section, we firstly introduce the experimental datasets, the state-of-the-art methods for comparison, evaluation measures, and parameter settings. Then we show the experimental results compared to these methods.

*5.1. Experimental Setup*

5.1.1. Datasets

We adopt four real-world short texts corpus from different domains. A brief description is as follows:

*News:* This dataset contains the titles of news collected from RSS feed of three websites (nyt.com, accessed on 5 September 2021, usatoday.com, accessed on 5 September 2021, reuters.com accessed on 5 September 2021). These short texts are categorized into seven classes: Sport, Business, U.S, Health, Sci&Tech, World, and Entertainment.

*Tweets:* This dataset contains tweets collected from Twitter. Each tweet has a hashtag and tweets with the same hashtag are considered to be in the same category. This corpus contains a total of 100 categories.

*Snippets:* This dataset contains the results of web search transaction [32]. Short texts are classified into eight categories: Business, Computers, Culture Arts, Education-Science, Engineering, Health, Politics-Society, and Sports.

*Captions:* This dataset contains captions solicited from Mechanical Turkers for photographs from Flickr and Pascal [33]. Captions are divided into 20 categories.

We pre-process these datasets by discarding stop-words and words that only occur once. The statistic information of four short texts datasets is shown in Table 2.

**Table 2.** Statistics of short texts datasets.

| DataSet | D | V | Len | C |
|---|---|---|---|---|
| News | 3000 | 10,906 | 12 | 7 |
| Tweets | 2500 | 4981 | 8 | 100 |
| Snippets | 3000 | 4705 | 14 | 8 |
| Captions | 4834 | 3165 | 5 | 20 |

Here column **D** means the number of short texts in datasets. Column **V** means the scale of the vocabulary. Column **Len** means the average length of short texts. Column **C** means the number of categories in datasets.

5.1.2. Methods

In this section, we introduce the state-of-the-art methods implemented for comparison.

*LDA [2]:* This model is one of the conventional topic models. We implement this method as the base method.

---

**Algorithm 1:** DESTM Sampling Process

---

**Input:** hyperparameter $\alpha$, $\beta$, $\gamma$, $\delta$

**Result:** topic associations $\vec{z}$

**1** initialize all variables;

**2** **while** *iteration* < *Maxiteration* **do**

**3**      **for** $i \in [1, N]$ **do**

```
// for each short text i, sample long document l*
// update counts of l* without text i
```

**4**          $n_{l*} = n_{l*} - 1$

**5**          sample long document $l^*$ according to Equation (5)

**6**          **if** $l_i^*$ *is a new long document* **then**

**7**              $L^* = L^* + 1$

**8**          **end**

```
// for each short text i, sample long document l
// update counts of l without text i
```

**9**          $n_l = n_l - 1$

**10**          $n_{l,t}^{-i} = n_{l,t}^{-i} - n_{i,t}$

**11**          $n_l^{-i} = n_l^{-i} - n_i$

**12**          **for** *short text i in* $l^*$ **do**

**13**              sample long document $l$ according to Equation (12)

**14**              **if** $l_i$ *is a new long document* **then**

**15**                  $L = L + 1$

**16**              **end**

**17**          **end**

```
// update counts of new sample l with text i
```

**18**          $n_l = n_l + 1$

**19**          $n_{l,t}^{-i} = n_{l,t}^{-i} + n_{i,t}$

**20**          $n_l^{-i} = n_l^{-i} + n_i$

```
// for each word in i, sample topic z
```

**21**          **for** $w \in [1, N_w]$ **do**

```
// update counts of z without word w
```

**22**              $n_{l,z} = n_{l,z} - 1$

**23**              $n_{w,z}^{-w} = n_{w,z}^{-w} - 1$

**24**              $n_z^{-w} = n_z^{-w} - 1$

**25**              sample topic $z$ according to Equation (16)

```
// update counts of new sample z with word w
```

**26**              $n_{l,z} = n_{l,z} + 1$

**27**              $n_{w,z}^{-w} = n_{w,z}^{-w} + 1$

**28**              $n_z^{-w} = n_z^{-w} + 1$

**29**          **end**

**30**      **end**

**31** **end**

---

*DSTM [18]:* This method incorporates the Spike and Slab prior that restricts the number of topics a short text corresponds to. Through this strategy, DSTM alleviates the problem of the sparsity of local word co-occurrence information.

*SATM [21]:* This method is the first self-aggregated topic model that aggregates short texts into latent long documents. This model is a two-phase model. The first phase generates latent long documents, and the second phase generates topics. Latent long documents can bring a lot of additional word co-occurrence information.

*PTM [22]:* This method is the state-of-the-art self-aggregated topic model. It generates latent long documents along with topics, which is simpler than model SATM. Additionally, aggregating short texts into long documents can overcome the problem of the sparsity of word co-occurrence information.

*SPTM [22]:* This method incorporates the Spike and Slab prior that restricts the number of topics a long document corresponds to. When the number of long documents is defined as too small, this method can avoid long documents becoming ambiguous.

### 5.1.3. Evaluation Measures

In this section, we introduce two measures: PMI score and classification accuracy.

### PMI Score

We calculate the point-wise mutual information (PMI) [34] score to evaluate the coherence of topics. We use the latest dump of Wikipedia articles as the auxiliary dataset to calculate the PMI score of words. This auxiliary dataset contains 5 million documents and 14 million words of vocabulary. We build a sliding window of 10 words and calculate the PMI score between any two words that appear in the sliding window. The equation of the PMI score is as follows:

$$PMI(w_i, w_j) = \log \frac{p(w_i, w_j)}{p(w_i)p(w_j)} \tag{17}$$

Here $w_i$ and $w_j$ are two words that appear in one sliding window. $p(w_i, w_j)$ is the joint probability of the word pair. $p(w_i)$ and $p(w_j)$ are marginal probabilities of words. Then, we generate top-N words with the highest probabilities correspond to a topic. Word pairs of top N words are generated and the average PMI score is calculated according to the PMI score from the Wikipedia dataset. Finally, we calculate the average PMI score of all topics.

### Classification Accuracy

We calculate the classification accuracy to evaluate the distributions between short texts and topics. Topics of a short text are suggested as features and the probabilities between a short text and its topics are suggested as feature values. Then we use the Support Vector Machine [35] as the classifier and calculate the classification accuracy according to these features of short texts through the bootstrap validation method [36]. The equation of the bootstrap accuracy is as follows:

$$ACC_{boot} = \frac{1}{b} \sum_{1}^{b} (0.632 * ACC_b + 0.368 * ACC_{total}) \tag{18}$$

For a short text set with $N$ short texts, we sample $N$ instances with replacement. Then we generate a SVM classifier from these instances and calculate the accuracy $ACC_b$ on the rest of the short texts that are not sampled. $ACC_{total}$ means the accuracy calculated on the total short texts set according to the classifier. This procedure will be repeated $b$ times and finally, the average accuracy $ACC_{boot}$ is calculated.

### 5.1.4. Parameter Settings

The Gibbs sampling method samples topics through thousands of iterations. For every model, We execute 3000 iterations and ignore first 1000 iterations to skip the convergence phase. For model LDA, we set $\alpha = 0.1$ and $\beta = 0.01$ according to paper [10]. For DSTM,

we set $\pi = 0.1$ and $\gamma = 0.01$ according to paper [22]. Researchers in this paper find these settings outperform settings in paper [18]. For SATM, we set the number of long documents as 300, $\alpha = 50/T$ and $\beta = 0.1$ according to settings in paper [21]. For PTM, we set the number of long documents as 1000, $\alpha = 0.1$ and $\beta = 0.01$ according to paper [22]. For SPTM, we set the number of long documents as 1000, $\alpha = 0.1$, $\beta = 0.01$, $\gamma_0 = 0.1$ and $\bar{a} = 10^{-12}$ according to paper [22]. The number of long documents is set to 1000 because this number is the appropriate number according to the scale of our four datasets. For our model (DESTM), we set $\alpha = 15$, $\beta = 15$, $\gamma = 0.1$, $\delta = 0.01$, and $\eta = 0.0$.

### 5.2. Topic Evaluation by Topic Coherence

In this section, we evaluate the topic coherence of the state-of-the-art models by calculating the PMI score. We generate 5, 10, and 15 topics for each model and choose the top 10 words for each topic according to the probability. The results are shown in Figure 2.

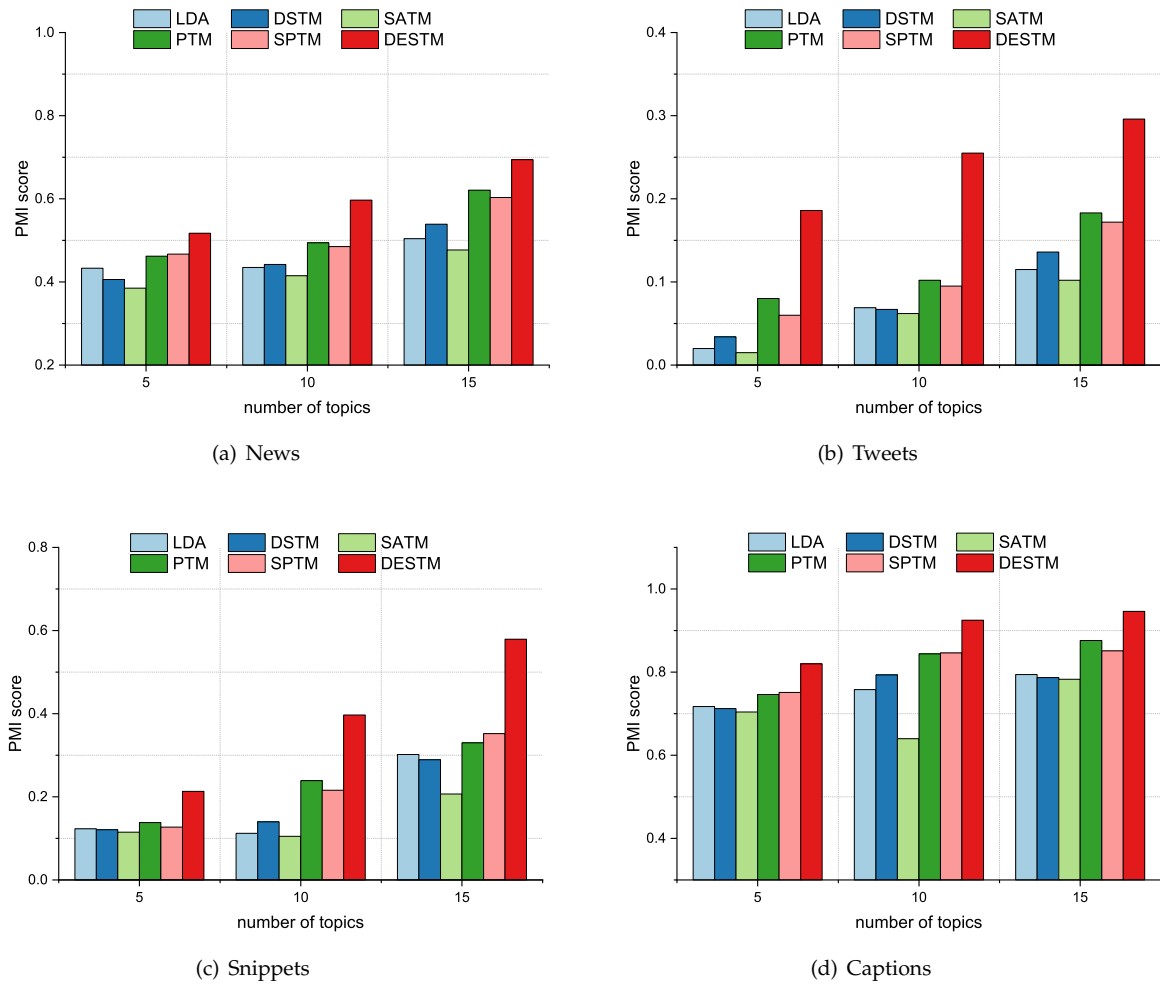

**Figure 2.** Average PMI score of all models on 4 datasets.

The results show that the LDA method achieves very poor performances on short texts. As a conventional method, LDA cannot overcome the problem caused by the sparsity of word co-occurrence information. The DSTM model achieves a little improvement compared to LDA. Although DSTM restricts the number of topics a short text corresponds to, this method cannot provide enough additional word co-occurrence information. So DSTM still suffers from the sparsity of word co-occurrence information. Surprisingly, the SATM model is worse than LDA in most cases. That is because SATM easily causes overfitting. The results of PTM and SPTM are the second or third best performances. The SPTM method can

only improve the performances when the number of long documents is defined too short. However, we set this number to be 1000 which is an appropriate number for the scale of our datasets. So the performances of PTM and SPTM are almost the same. The results of these two models show that aggregating short texts into latent long documents can overcome the sparsity of word co-occurrence information. However, non-semantic word co-occurrence information restricts their performances. Our method (DESTM) successfully aggregates semantically related short texts together according to document embeddings and produces sufficient and semantically related additional word co-occurrence information. So the results show that DESTM achieves a significant improvement and outperforms the other state-of-the-art methods.

### 5.3. Topic Evaluation by Classification Accuracy

In this section, we evaluate the probability relations between short texts and topics. We also generate 5, 10, and 15 topics on 4 datasets. The results are shown in Figure 3.

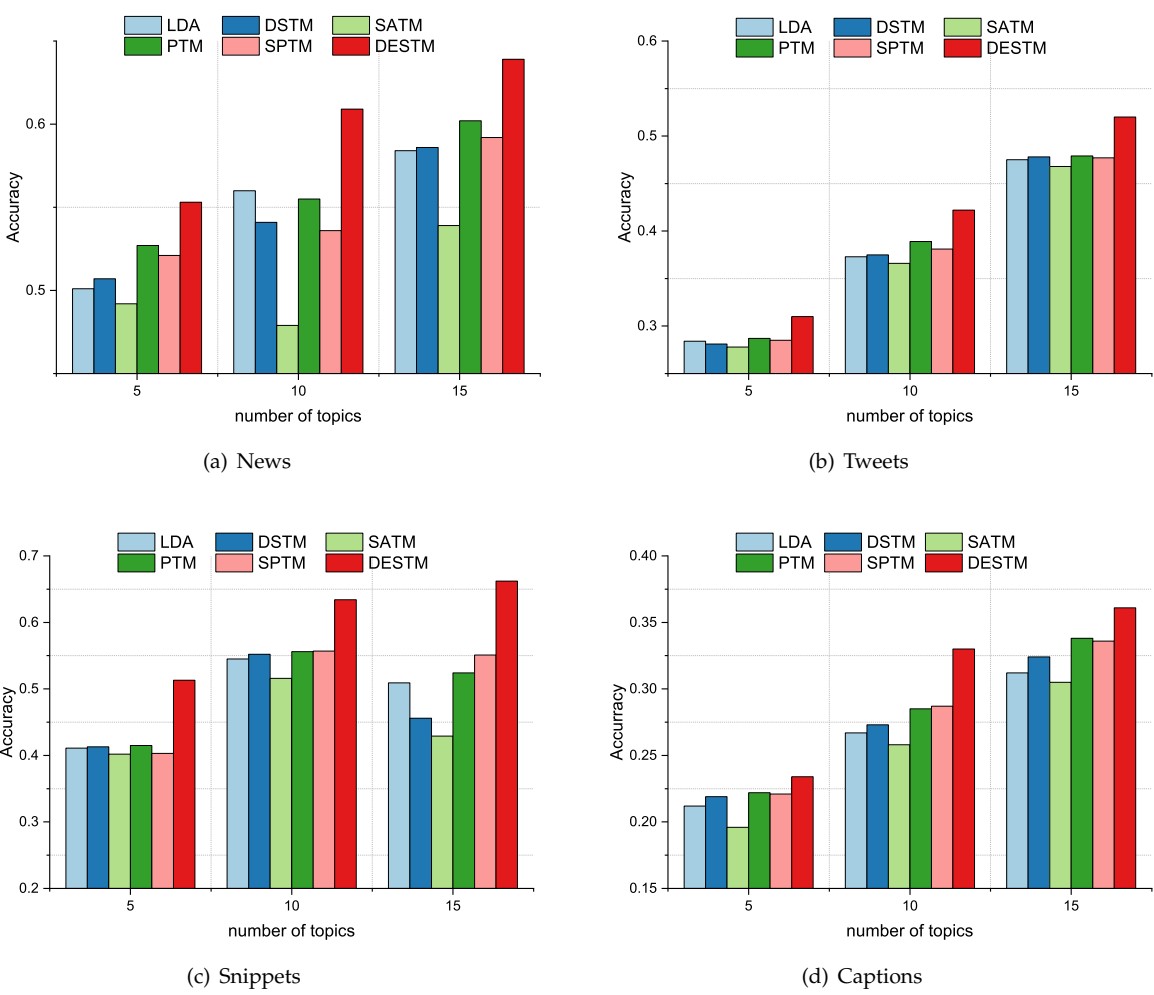

**Figure 3.** Classification accuracy of all models on 4 datasets.

In these results, the LDA model achieves poor performances as it cannot overcome the sparsity of word co-occurrence information. DSTM performs better than LDA. Restricting the number of topics a short text corresponds to can relieve the sparsity of local word co-occurrence information for each topic. However, without sufficient word co-occurrence information, the DSTM method cannot achieve a great improvement. SATM also performs the worst because of the overfitting problem. PTM and SPTM achieve the second or third best results in most cases. Latent long documents in these two models provide sufficient

word co-occurrence information and achieve an improvement on short texts. However, PTM and SPTM only achieve little improvement. In these two models, the probabilities between short texts and topics are generated from the distribution $P(z|d)$. This distribution is generated according to the equation $P(z|d) = P(z|l)P(l|d)$. $z$ means variables of topics, $d$ means variables of short texts, and $l$ means variables of the long documents. PTM and SPTM cannot avoid aggregating semantically irrelevant short texts together and these short texts will bring noisy information into long documents. So the noisy information in long documents will directly influence the probability between topics and short texts and leads to poor performances on Tweets and Snippets. For our model, the performances are superior compare to the other state-of-the-art models on four datasets. The results show that long documents generated by our model are more reasonable and contain less noisy information.

*5.4. Experimental Results for Complete and Partial Document Embeddings*

In this section, we discuss the impact of noisy information of document embeddings in our model. We compared several DESTM models with different parameter settings on dataset News. Firstly we use the PTM model to represent the condition without any document embeddings. Settings of PTM follow experiments above. Secondly, we set parameters of DESTM as $\alpha = 0$ and $\eta = -1$. The model with these settings means aggregating short texts according to the complete document embeddings and of cause including all noisy information. Thirdly, we set $\alpha = 15$ and $\eta = -1$. The model with these settings means aggregating short texts according to the similarity distribution and distances of short texts without discarding any distances. Then we vary the threshold $\eta$ to be $-0.5$, $0.0$, and $0.5$. We get models incorporating partial document embeddings with different threshold values. Finally, we set $\alpha = 15$ and $\eta = 1$. The model with these parameters means aggregating short texts only according to the similarity distribution. For other parameters of DESTM, we set $\beta = 2$, $\gamma = 0.1$, and $\delta = 0.01$. The evaluation results of PMI score and classification accuracy are shown in Figure 4.

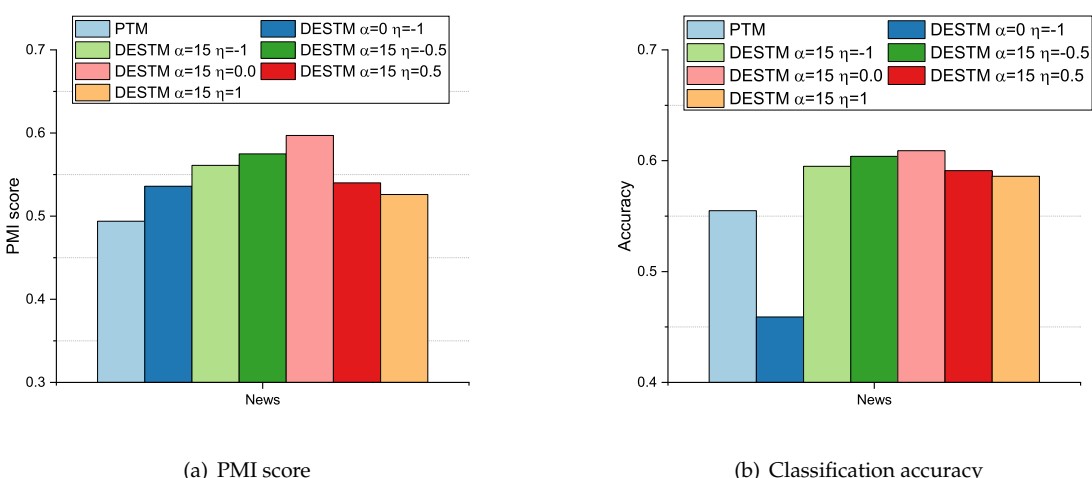

(a) PMI score          (b) Classification accuracy

**Figure 4.** Comparison results between models with complete and partial document embeddings.

In these results, the PTM model achieves the worst PMI score and the second-worst classification accuracy. Without any additional semantic information, the performances of the PTM model are very poor. The DESTM model with $\alpha = 0$ and $\eta = -1$ incorporates the complete document embeddings without discarding noisy information. The model of these settings achieves the third-worst PMI score and the worst classification accuracy. Results show that aggregating short texts directly according to document embeddings will incorporate a lot of noisy information and leads to poor performances. For models with $\alpha = 15$ and $\eta$ varies from -1 to 1, the number of distances between short texts that

are discarded varies from zero to all distances. The results show that the model with $\eta = 0.0$ achieves the best performance. The model with $\eta = -0.5$ achieves the second-best performance. The model with $\eta = -1$ is the third best and the model with $\eta = 0.5$ is the fourth one. Finally, the model with $\eta = 1$ achieves the second-worst PMI score and the third-worst classification accuracy. The results show that similarity distributions and distances of short texts are all beneficial to our model and both information should be retained. The threshold of distances should be carefully designed, discarding too many distances or retaining too many distances will all reduce the performances.

### 5.5. Semantic Explanations of Topic Demonstrations

Topics are constructed by words and should be understandable by human beings. In this section, we analyze the semantic meanings of topics. We generated 10 topics with the top 10 words on the News dataset. Results are shown in Table 3.

**Table 3.** Top 10 words for the topic News.

| Topic | Top-10 Words |
| --- | --- |
| Topic1 | game coach season year scored night team big points boston |
| Topic2 | billion group bank financial deal company international chief fund companies |
| Topic3 | oil prices year percent high rose growth economic demand fell |
| Topic4 | japan nuclear power health study earthquake plant government crisis world |
| Topic5 | air officials libyan city nato muammar country people military killed |
| Topic6 | open world final french year champion championship week players national |
| Topic7 | president court state obama federal minister judge prime year election |
| Topic8 | american long online video apple weekend internet music million year |
| Topic9 | president al killed security country forces government people bin laden |
| Topic10 | show life years news tv star play broadway wedding making |

We can interpret these topics into semantic meanings. Topic1 is about sports news of basketball. Topic2 is about financial news. Topic3 is about economic news of oil prices. Topic4 is about disaster news in the world. Topic5 is news about international politics. Topic6 is also sports news but corresponds to tennis. Topic7 is political news of the election. Topic8 is about technological news of multimedia service. Topic9 is political news about terrorists. Topic10 is about entertainment news. Results show that we can understand the semantic meanings of topics generated by our model.

### 5.6. Efficiency Analysis

In this section, we evaluate the efficiency of these models. All topic models are implemented by java with the java 8 virtual machine and are calculated on the same computer. The hardware environment of the experimental computer is a CPU of i5 8400 and an 8G memory. We generate 10 topics on Snippets dataset and the results of initialization time and average time of each iteration are shown in Table 4. The unit of time is milliseconds.

**Table 4.** Time of initialization and average time of each iteration.

| Models | Time of Initialization (ms) | Average Time per Iteration (ms) |
| --- | --- | --- |
| LDA | 15 | 15 |
| DSTM | 71 | 2631 |
| SATM | 16 | 815 |
| PTM | 15 | 193 |
| SPTM | 16 | 237 |
| DESTM | 310 | 442 |

In this result, the conventional model LDA is the fastest because this model is the simplest model. The DSTM model is the most inefficient model because it needs to sample

many prior variables that restrict the number of topics. SATM is the second most inefficient model because this model has two phases to calculate. The efficiency of PTM and SPTM is similar. PTM only needs sample long document variables and topics. SPTM is slower than PTM because it needs to sample additional topic selector variables. Our model, DESTM, is faster than DSTM and SATM, but slower compared to PTM and SPTM. That is because we need to sample additional long documents in the first step of the nested Chinese restaurant process. For each iteration of Gibbs sampling, our model will sample three latent variables. For each short text, we sample two latent long document variables, and for each word, we sample the latent topic variable. The time complexity of our model is $O(K^* + K + T)$. Here $K^*$ is the size of the total long document samples space in the first step of the nested Chinese restaurant process. We calculate $K^*$ as $K^* = \sum_{i=1}^{N} K_i^*$. Here $N$ means the number of short texts. $K_i^*$ means the size of the long document samples space in the first step of short text $i$. $K$ is the size of the total long document samples space in the second step of the nested Chinese restaurant process. We calculate $K$ as $K = \sum_{i=1}^{N} K_i$. $K_i$ means the size of the long document samples space in the second step of short text $i$. $T$ is the size of the total topic samples space. $T$ is calculated as $T = NT^*$. Here $T^*$ is the size of the topic samples space of each short text.

## 6. Conclusions

In this paper, we propose a self-aggregated topic model incorporating document embeddings (DESTM). We calculate document embeddings from short texts and incorporate this information to aggregate semantically related short texts together and to provide sufficient additional semantically related word co-occurrence information. To discard noisy information generated by the sparse word co-occurrence information, document embeddings are separated into global and local semantic information. The global semantic information is the similarity distribution on the entire dataset, and the local semantic information is the distances between similar short texts. Then we use the nested Chinese restaurant process to incorporate these two kinds of semantic information. The global information is incorporated through the first step of the nested Chinese restaurant and the local information is incorporated through the second step. The experimental results show that our model outperforms the other state-of-the-art models in terms of PMI score and classification accuracy. However, our model still has a limitation. When document embeddings that contain noisy information are discarded, we cannot avoid discarding other useful information in these document embeddings like context semantic information. So semantic information incorporated into our model is not sufficient. In the future, we will try to incorporate other kinds of semantic information such as the global context semantic information to provide more sufficient semantic information.

**Author Contributions:** Conceptualization, Y.N.; methodology, Y.N.; software, Y.N.; validation, Y.N.; formal analysis, Y.N.; investigation, Y.N.; resources, Y.N.; data curation, Y.N.; writing—original draft preparation, Y.N.; writing—review and editing, H.Z.; visualization, Y.N.; supervision, J.L.; project administration, J.L.; funding acquisition, J.L. All authors have read and agreed to the published version of the manuscript.

**Funding:** This research received no external funding.

**Institutional Review Board Statement:** Not applicable.

**Informed Consent Statement:** Not applicable.

**Data Availability Statement:** Datasets can be obtained from https://github.com/overlook2021/short-texts.git accessed on 5 September 2021.

**Conflicts of Interest:** The authors declare no conflict of interest.

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
