# Peer review of "A Nested Chinese Restaurant Topic Model for Short Texts with Document Embeddings"

_applsci, doi:10.3390/app11188708_

Round 1

Reviewer 1 Report

  1. Related works section shows two main strategies, i.e., "Models with auxiliary information" and "Models without auxiliary information", respectively. I suggest adding a figure or table to clearly compare the similarities and differences between these two strategies in the related work section. Such table or figure may be especially helpful for any readers to understand the proposed method.
  2. At line 400, the detailed formula for the classification accuracy is required.
  3. In Figure 4, what is the meaning of "PTM" on its y-axes? Does it indicate "PMI"? I guess that the y-axes in Figure 4 indicates "PMI score" not "PTM". Please check this.
  4. In Table 3, its caption "efficiency" is not enough to show what the table shows. The table describes the average execution time of the compared methods. Then, the caption of this table should explain it clearly. However, the current caption is not clear at all. Thus, you should correct this caption.
  5. At line 519, the time complexity of the proposed method is O(K* + K + T). How was this complexity derived? The detailed explanations about this should be enhanced in the "Efficiency Analysis" section. 
  6. What is the shortcoming of this study? It is good to briefly address the weakness (or limitation) of this study in the conclusion section.
  7. More latest reference papers related to this study are required in the reference list (published between 2020 and 2021).

Reviewer 2 Report

The article presents a modeling technique for the semantic interpretation of short texts that is based on the nested Chinese restaurant model. It allows for the incorporation of both global and local semantics for improved interpretation over time.

The problem of correct semantic classification of short texts is indeed an important one in particular in the age of social media messaging, fake news and spam. While the idea of using Nested Chinese Restaurant Topic Models in document understanding is not new its application in the particular context of short texts appears novel and produces good results. The paper presents an introduction to the background material with a good discussion of state-of-the-art and related work. The new model is well described and reproducible for the readers and the extensive experimental testing and presentation of results round out the contributions of the paper.

While the paper is overall well-structured and generally well-written, it still suffers from some minor presentational issues:

  • Algorithm 1 would benefit from some additional explanation (e.g. in the form of interspersed comments).
  • The paper also suffers from poor visual quality of the mathematics presentation. I assume this is due to the authors' choice of word processing over a high quality TeX typesetting engine.
  • Finally, the paper needs thorough proof reading for English mistakes. Below are some of the problems from the introduction alone. Many more exist throughout the paper:

29:  provide poor performances: bad grammar
33:  is incorporating > is to incorporate

53: letting local word co-occurrence no longer sparse. ?? Leading to local word
    co-occurrence to no longer being sparse??
59: exits -> exist
68: word co-occurrence(s ?).
70: brought >on< by sparse word co-occurrence. (or better resulting from)
83: The sparse of word co-occurrence  -> The sparseness ... ?
87: At the first step -> In the first step
88: ditto

Round 2

Reviewer 1 Report

I have reviewed this manuscript and found that it has been carefully revised. All issues mentioned in the first review have been addressed in this manuscript. Thus, I suggest to finally check any grammatical errors or typos before submitting this manuscript.